# Mixed-Membership Community Detection
# via Line Graph Curvature

**Yu Tian**                                                       YU.TIAN@MATHS.OX.AC.UK
*University of Oxford*

**Zachary Lubberts**                                             ZLUBBER1@JHU.EDU
*Johns Hopkins University*

**Melanie Weber**                                               MWEBER@SEAS.HARVARD.EDU
*Harvard University**

**Editors:** Sophia Sanborn, Christian Shewmake, Simone Azeglio, Arianna Di Bernardo, Nina Miolane

## Abstract

Community detection is a classical method for understanding the structure of relational data. In this paper, we study the problem of identifying mixed-membership community structure. We argue that it is beneficial to perform this task on the *line graph*, which can be constructed from an input graph by encoding the relationship between its edges. Here, we propose a curvature-based algorithm for mixed-membership community detection on the line graph. Our algorithm implements a discrete Ricci curvature flow under which the edge weights of a graph evolve to reveal its community structure. We demonstrate the performance of our approach in a series of benchmark experiments.

**Keywords:** Line Graph, Mixed-membership community detection, Ollivier-Ricci curvature, Curvature-based Network Analysis

## 1. Introduction

Community detection is of central importance to the study of relational data, such as graphs or networks. It seeks to identify clusters or densely interconnected substructures in a given graph. Such structure is ubiquitous in relational data: We may think of friend circles in social networks, pathways in biochemical networks or article categories in Wikipedia. A standard mathematical model for graphs with a community structure is the *Stochastic Block Model (SBM)*, which has led to fundamental insights into the detectability of communities (Abbe, 2017).

Classically, communities are identified by clustering the nodes of the graph. Popular methods include the Louvain algorithm (Blondel et al., 2008), the Girvan-Newman algorithm (Girvan and Newman, 2002) and Spectral clustering (Cheeger, 1969; Fiedler, 1973; Spielman and Teng, 1996). Recently, there has been growing interest in another class of algorithms, which apply a geometric lens to community detection. *Curvature-based community detection* seeks to understand the structure of complex networks through a characterization of their geometry. Curvature is a classical tool in Differential Geometry, which is used to characterize the local and global properties of geodesic spaces. While originally defined in continuous spaces, discrete notions of curvature have recently seen a surge of interest. Of

---

* Work partially done while at the University of Oxford.

particular interest are discretizations of Ricci curvature, which is a local notion of curvature that relates to the volume growth rate of the unit ball in the space of interest (geodesic dispersion). Curvature-based community detection utilizes the observation that edges between communities (so called *bridges*) have low Ricci curvature. By identifying such edges, we can learn a partition of the graph into its communities.

Much of the existing literature focuses on communities with unique membership, i.e., each node can belong to one community only. However, in many complex systems that generate relational data, we find mixed-membership structures: A member of a social network might belong to a circle of high school friends and a circle of college friends, a protein may have different functional roles in a biochemical network. Consequently, a growing body of literature studies the detection of overlapping communities (Airoldi et al., 2008; Yang and Leskovec, 2013; Zhang et al., 2020). Observe that the mixed-membership structure precludes the existence of bridges between communities, because the communities overlap (see also Fig. 1(left)). This renders community detection methods that rely on graph partitions, such as the curvature-based methods discussed above, inapplicable.

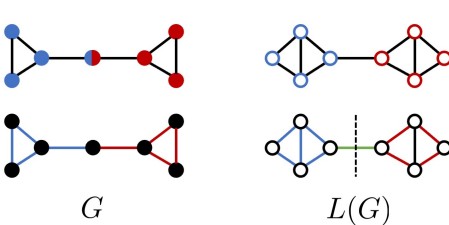

Figure 1: Community structure in a graph $G$ and its line graph $L(G)$ with node labels (top) and edge labels (bottom).

In this paper, we propose a curvature-based algorithm for detecting mixed-membership community structure. We show that while partition-based approaches do not recover the natural communities when applied to the original graph, they perform well on the line graph. The *line graph* encodes the connectivity between the edges of the graph, rather than the nodes. This reformulation of the relational information in the graph allows for disentangling the overlapping community structure. Our proposed curvature-based detection method then identifies edges in the line graph, which can be cut to partition the graph into its communities (see Fig. 1(right)).

### 1.1. Summary of Contributions

The two main contributions of this paper are as follows:

1. We demonstrate that it is beneficial to perform mixed-membership community detection on the line graph, rather than the original graph. This is of particular importance for classical partition-based methods that seek to identify communities and cut edges between them, called *bridges*. Such bridges do not exist between overlapping communities in the original graph, but can be found in the line graph.

2. We propose a curvature-based method that learns an overlapping community structure by evolving edge weights in the line graph under a Ricci curvature flow (Algorithm 4.1), reinforcing the meso-scale structure of the network from which the community structure can be inferred.

## 1.2. Related work

Mixed-membership community detection is widely studied in the Network Science and Data Mining communities. Notable approaches include Bayesian methods (Airoldi et al., 2008; Hopkins and Steurer, 2017), matrix factorization (Yang and Leskovec, 2013), spectral clustering (Zhang et al., 2020), and vertex hunting (Jin et al., 2017), among others. In addition to the mixed-membership model that we consider here (Airoldi et al., 2008), there is a significant body of literature on closely related overlapping community models (Lancichinetti et al., 2009; Xie et al., 2013), which also study the problem of learning non-unique node labels.

Curvature-based community detection methods for non-overlapping communities have recently received growing interest (Ni et al., 2019; Sia et al., 2019; Weber et al., 2018). Such approaches utilize notions of discrete curvature (Ollivier, 2010, 2009; Forman, 2003) to partition networks into communities, based on the observation that edges between communities have low curvature. The absence of such bridges in overlapping communities renders these approaches inapplicable to the setting studied in this paper. To the best of our knowledge, our algorithm is the first to study mixed-membership community structure with curvature-based methods. More generally, curvature-based network analysis (see, e.g., (Weber et al., 2017a,b)) has been applied in many domains, including to biological (Elumalai et al., 2022; Weber et al., 2017c; Tannenbaum et al., 2015), chemical (Leal et al., 2021; Saucan et al., 2018), social (Painter et al., 2019) and financial networks (Sandhu et al., 2016). Community detection and more generally network analysis via the line graph has been recently studied in (Chen et al., 2017; Krzakala et al., 2013; Lubberts et al., 2021; Evans and Lambiotte, 2010).

## 2. Line Graph Curvature

We begin by introducing the notion of a line graph, followed by a definition of a notion of discrete Ricci curvature for line graphs. We endow graphs with the usual *path distance*.

## 2.1. Line Graphs

Consider an unweighted, undirected graph $G = (V, E)$ (the *original graph*), with node set $V$ and edge set $E \subseteq \binom{V}{2} = \{\{u, v\} : u, v \in V\}$. Its *line graph* is given by the relationship of its edges, in the sense that an edge in the line graph represents adjacent edges in the original graph. Formally, let $L(G) = (E, \mathcal{E})$ denote the line graph of $G$, where $\mathcal{E}$ is given by all pairs of adjacent edges $\{\{u, v\}, \{r, s\}\} \in \mathcal{E}$, i.e., for which $|\{u, v\} \cap \{r, s\}| = 1$.

Of particular interest for this work is the relationship between the structural properties of the original graph and the line graph. It is easy to see that the following relation of node degrees holds:

$$\deg(\{u, v\}) = \deg(u) + \deg(v) - 2 \,. \tag{2.1}$$

The equation relates the degree of a node in the line graph ($\{u, v\} \in L(G)$) to that of the adjacent nodes in the original graph ($u, v \in G$).

## 2.2. Discrete Ricci Curvature of Line Graphs

Our notion of discrete curvature relates geodesic dispersion to optimal mass transport. In particular, we consider the transportation cost between two distance balls (i.e., node neighborhoods) along an edge in the network. Formally, for an edge $e = \{v_1, v_2\}$, we endow the neighborhoods of each of its nodes with a uniform measure, i.e.,

$$m_{v_1}(u) := \frac{1}{\deg(v_1)} \qquad \forall u, \ s.t. \ d(u, v_1) = 1$$

$$m_{v_2}(u) := \frac{1}{\deg(v_2)} \qquad \forall u, \ s.t. \ d(u, v_2) = 1 .$$

We then define Ollivier's Ricci curvature (Ollivier, 2010) (ORC) with respect to the Wasserstein distance $W_1$ between those measures, i.e.,

$$\kappa_{v_1 v_2} := 1 - W_1(m_{v_1}, m_{v_2}) . \tag{2.2}$$

Recall that the Wasserstein distance between measures $m_{v_1}, m_{v_2}$ is given by

$$W_1(m_{v_1}, m_{v_2}) = \inf_{m \in \Gamma(m_{v_1}, m_{v_2})} \int_{(u, u') \in V \times V} d(u, u') m(u, u') \, du \, du' . \tag{2.3}$$

Here, $\Gamma(m_{v_1}, m_{v_2})$ denotes the set of all measures between pairs of nodes $V \times V$ with marginals $m_{v_1}, m_{v_2}$. Fig. 2 analyzes ORC in a sample SBM.

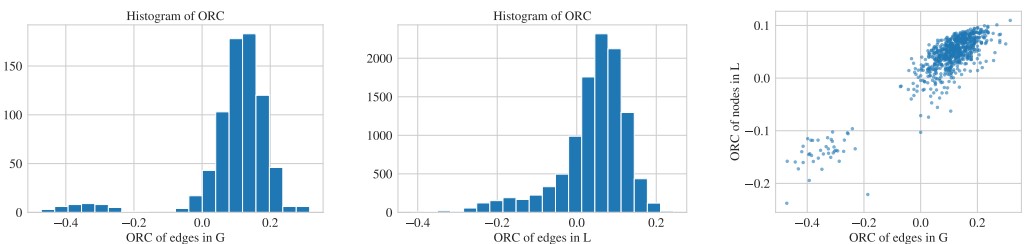

Figure 2: ORC of edges in $G$ (left) and of edges in its line graph $L$ (middle) for a two-block planted SBM of size $n = 100$ with intra-community edge probability $p_{in} = 0.3$, inter-community edge probability $p_{out} = 0.02$. The plot on the right shows the correlation between edge curvatures in $G$ and $L(G)$.

Curvature-based community detection, which will be the focus of this paper, relies on the observation that *bridges* between communities are more negatively curved than edges within communities. We give some intuition for this observation: ORC can be linked to the behavior of two random walks starting at adjacent nodes (Ollivier, 2009; Jost and Liu, 2014). Informally, they are more likely to draw apart if the edge connecting the nodes has negative ORC, and to draw closer together otherwise. When the random walks start at nodes adjacent to a bridge, they typically proceed into the communities to which the respective nodes belong, and consequently draw apart. On the other hand, random walks starting at nodes adjacent to an internal edge are more likely to stay nearby, as they remain within the community. Hence, we expect bridges to be much more negatively curved than internal edges, which is easily confirmed empirically.

## 3. Mixed-membership Community Detection on the Line Graph

Community structure is a hallmark feature of networks, characterized by clusters of nodes that have more internal connections than connections to nodes in other clusters. The focus of this paper is on networks with mixed-membership, which quantifies the idea of overlapping communities, so that nodes may belong to more than one cluster.

**Definition 1 (Mixed-Membership SBM)** *A* Mixed-Membership Stochastic Block Model (MMB) *(Airoldi et al., 2008), can be seen as an extension of the classical SBM model. In both models, each element of the adjacency matrix* $\mathbf{A}$ *above the diagonal is an independent Bernoulli random variable whose expectation only depends on the block memberships of the corresponding nodes. In the MMB, it is possible that nodes belong to more than one block, with various affiliation strengths. Specifically, we assume that the expectation* $\mathbb{E}[\mathbf{A}]$ *has the form* $\mathbb{E}[\mathbf{A}] = \mathbf{X}\mathbf{B}\mathbf{X}^T$, *where* $\mathbf{X} \in [0,1]^{n \times k}$ *is the community membership matrix with* $X_{il}$ *indicating the affiliation of node* $i$ *with community* $l$ *and* $\sum_l X_{il} = 1$, *and* $n$ *denotes the size of the network. The matrix* $\mathbf{B} \in [0,1]^{k \times k}$ *encodes the block connection probabilities. If* $X_{il} = 1$ *for some* $l$, *we call vertex* $i$ *a* pure node. *When all nodes are pure, we recover the ordinary SBM model. In our experiments, we consider a planted version where* $\mathbf{B}_{ii} = p_{in}$, $\forall i$, *and* $\mathbf{B}_{ij} = p_{out}$, $\forall i \neq j$.

Of particular importance for classical community detection are edges between clusters, often called *bridges*. This includes many popular partition-based approaches (Blondel et al., 2008; Girvan and Newman, 2002; von Luxburg, 2007), which rely on identifying and cutting such bridges to partition the graph into communities. However, overlaps between communities preclude the existence of these bridges, which limits the applicability of partition-based approaches. In this paper we argue that the line graph provides a natural input for partition-based mixed-membership community detection. While nodes may not have a unique label in this model, the adjacent edges may still be internal, connecting two nodes that are in the same community (at least partially). In this case, each edge is associated with a single community (see Fig. 1(left)). Consequently, representing the relationships among edges in a line graph, we disentangle the overlapping communities. Each edge in the line graph appears at a vertex in the original graph. When the vertex has mixed membership, a bridge between the communities arises in the line graph (Fig. 1(right)). In Appendix A, we illustrate this observation on two small sample networks. Notice that the line graph is typically larger in size than the original graph. This necessitates special attention to the scalability of the algorithm. In particular, in a graph with $n$ vertices and average degree $n\alpha_n$, we expect to have $n^2\alpha_n$ edges. In the line graph, this means $n^2\alpha_n$ vertices and $n^3\alpha_n^2$ edges, though this number increases with greater variation of the vertex degrees in the original graph. The much larger size of the line graph is also evident in our experiments below, see Table 2. In the next section, we will discuss a scalable approach that utilizes discrete Ricci curvature to identify bridges in the line graph.

---

**Algorithm 1** ORC Ricci flow on the Line Graph

---

**Input:** Graph $G$, threshold $\Delta$.
Construct line graph $L(G) =: L^0$.
**for** $t = 0, \ldots, T-1$ **do**
> Compute ORC $\{\kappa_{uv}^t\}_{uv \in \mathcal{E}}$ for all edges in the line graph.
> Evolve edges under Ricci flow: $w_{uv}^t \leftarrow (1 - \kappa_{uv}^t)d_{uv}^t$.
> Renormalize edge weights: $w_{uv}^t \leftarrow \frac{|\mathcal{E}|d_{uv}^t}{\sum_{\{u',v'\} \in \mathcal{E}} d_{u'v'}^t}$.
> Cut edges with weight below threshold ($w_{uv}^t < \Delta$), resulting in new graph $L^{t+1}$.

**end**
**Output:** $L^T$.

---

## 4. Curvature-based Approach

### 4.1. Algorithm

In the community detection literature, approaches based on discrete Ricci curvature have been studied for unique-membership communities (Ni et al., 2019; Sia et al., 2019; Weber et al., 2018). Here, we propose a curvature-based method for *mixed-membership community detection*.

Like previous approaches that utilize Ollivier's notion of Ricci curvature (e.g., Ni et al. (2019)), we build on a notion of discrete Ricci flow first proposed by Ollivier (2009):

$$\frac{d}{dt}d_{uv}(t) = -\kappa_{uv}(t)d_{uv}(t) \qquad (\{u,v\} \in \mathcal{E}) . \tag{4.1}$$

In our approach, $d_{uv}$ denotes the path distance between adjacent nodes $u, v \in E$ in the line graph and $\kappa_{uv}$ Ollivier's Ricci curvature along that edge. The key idea of the community detection approach is to consider a family of weighted graphs $L^t = \{E, \mathcal{E}, w^t\}$ for an input graph $G$ ($L^0 := L(G)$), which is constructed by evolving the edge weights under Ricci flow, i.e.,

$$w_{uv}^t \leftarrow (1 - \kappa_{uv}^t)d_{uv}^t \qquad (\{u,v\} \in \mathcal{E}) , \tag{4.2}$$

where the curvature $\kappa_{uv}^t$ and path distance $d_{uv}^t$ is computed on the graph $L^t$. The procedure is initialized with the unweighted line graph $L(G) =: L^0$ contructed from the input, i.e., $w_{uv}^0 = 1$ for all edges $\{u,v\} \in \mathcal{E}$. In each iteration, the edge weights are renormalized using

$$w_{uv}^t \leftarrow \frac{|\mathcal{E}|d_{uv}^t}{\sum_{\{u',v'\} \in \mathcal{E}} d_{u'v'}^t} . \tag{4.3}$$

Over time, the negative curvature of edges that bridge communities becomes stronger, since edges with a lower Ricci curvature contract slower under Ricci flow. On the other hand, edges with higher Ricci curvature contract faster. This results in a decrease of the weight of internal edges over time, while the weight of the bridges increases (see Fig. 3). With that, the discrete Ricci flow reinforces the meso-scale structure of the network. Our approach is schematically shown in Alg. 4.1.

Applying Alg. 4.1 has a *coarsening* effect: In each iteration, we remove edges with weights below a predefined threshold $\Delta$. While bridges are preserved under the Ricci flow,

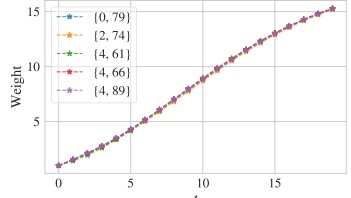

Figure 3: We consider a two-block planted SBM of size $n = 100$ with $p_{in} = 0.3$ and $p_{out} = 0.02$ (as in Fig. 2), visualized in the left panel. We select representative edges within a community, shown in orange on the graph, and plot the evolution of their weights in the middle panel. In the right panel, we show this evolution for selected edges between communities (i.e., bridges), shown in blue on the graph.

edges with less structural importance (such as internal edges) are removed successively as their weight decreases. The sparsified graph retains the meso-scale structure of the original graph. In particular, its community structure is preserved.

In order to recover the community structure in the original graph, we cut the edges in the line graph with the highest weight (or, equivalently, the most negative curvature) after evolving edge weights for $T$ iterations. The resulting partition of the line graph delivers an edge clustering from which we infer the community labels: We obtain a mixed-membership label vector $y$ for each node $v$ by computing

$$y_l(v) = \frac{1}{|E_v|} \sum_{e \in E_v} \chi_l(e) \,, \tag{4.4}$$

where $\chi_l$ is the indicator for the edge cluster $C_l$. Intuitively, each edge belonging to cluster $l$ that is incident on a node $v$ adds more evidence of affinity between the node $v$ and the cluster $C_l$.

### 4.2. Experiments

To measure the performance of the algorithm, we use the *normalised mutual information (NMI)* that has been extended to the setting of overlapping communities (Lancichinetti et al., 2009), one of the commonly used measures in evaluating the quality of the detected partitions (Xie et al., 2013). A detailed definition of the NMI can be found in Appendix B.

Alg. 4.1 assumes that the edge weight threshold $\Delta$ and the stopping time $T$ are given at initialization. In the experiments we use modularity as a heuristic for choosing $\Delta$ and manually fine tune $T$. We leave a systematic study of hyperparameter choices to future work.

### 4.2.1. SYNTHETIC DATA

For the first set of experiments, we generate graphs with $p_{in} = d/n$, where the expected mean degree is chosen to be $d = 30$, network size $n = 300$, and we have $k = 2$ communities. We consider different probabilities $\pi_o$ for mixed-membership of a node, ranging from 0.005

to 0.03. Pure nodes are evenly distributed between the two communities. We vary the ratio of connection probabilities within and across communities $\rho = p_{out}/p_{in}$ from 0 up to 0.1. For each set of parameters, we generate $n_s = 10$ graphs and run our algorithm, reporting the average (mean) NMI, and standard deviation (SD). To maintain the detectability of the communities, we require the generated graphs to have modularity greater than 0.4 with the ground-truth communities (or 0.35 when $\rho = 0.1$ and $\pi_o \geq 0.025$, since modularity over 0.4 is rare in these cases). Our experimental results (see Tab. 1) demonstrate that the ORC approach successfully recovers mixed-membership community structure, with an NMI above 0.8 as we vary $\rho$ and $\pi_o$. As one may expect, the performance of our approach drops as the probability of intra-community edges or the overlap between communities increase.

| $\rho \backslash \pi_o$ | 0.005 | 0.01 | 0.015 | 0.02 | 0.025 | 0.03 |
|---|---|---|---|---|---|---|
| 0 | 0.99 | 0.98 | 0.98 | 0.96 | 0.95 | 0.95 |
| | $(3.3 \times 10^{-3})$ | (0.01) | (0.01) | (0.05) | (0.04) | (0.04) |
| 0.05 | 0.99 | 0.98 | 0.97 | 0.91 | 0.93 | 0.92 |
| | $(6.3 \times 10^{-3})$ | (0.01) | (0.01) | (0.09) | (0.07) | (0.09) |
| 0.1 | 0.97 | 0.96 | 0.94 | 0.87 | 0.91 | 0.89 |
| | (0.02) | (0.01) | (0.04) | (0.08) | (0.07) | (0.03) |

Table 1: Mean (SD) of NMI for the proposed mixed-membership community detection approach via ORC (Alg. 4.1) on a set of mixed-membership SBMs, with varying probability of mixed-membership nodes $\pi_o$ and different ratios of edges probabilities between and within communities $\rho$.

### 4.2.2. Real-world data

To demonstrate the performance of our algorithm on real data, we consider two data sets, (i) collaboration networks from DBLP (Yang and Leskovec, 2015), and (ii) ego-networks from Facebook (Leskovec and Mcauley, 2012), for which ground truth labels are available through SNAP (Leskovec and Krevl, 2014). The collaboration network is constructed by a comprehensive list of research papers in computer science provided by the DBLP computer science bibliography. Here, an edge from one author to another indicates that they have published at least one joint paper. There are intrinsic communities defined by the publication venue, e.g. journals or conferences. Here, we maintain the choice of two communities, and randomly select two venues whose sizes are over $n = 100$ and have at least one overlapping node. We report results for two networks: "DBLP-1" from publication venues no. 1347 and 1892, and "DBLP-2" from publication venues no. 1347 and 2459. In the second data set (ego-networks), all the nodes are friends of one central user, and the friendship circles set by this user can be used as ground truth communities. We carried out the preprocessing in a similar manner as Zhang et al. (2020), and then select two networks for which the modularity of the ground-truth communities is greater than 0.5: "FB-1" for no. 414, and "FB-2" for no. 1684. To better understand the characteristics of the different real networks, we provide the following summary statistics for each network (see Table 2): (i) average node degree $d$, (ii) degree heterogeneity measured by the standard deviation of

node degrees over $d$, (iii) the actual proportion of overlapping nodes, and (iv) the modularity of the ground-truth community. We note that the Facebook networks tend to be denser, with more overlapping nodes, while the collaboration networks tend to have more heterogeneous degrees.

| | $n$ | $|E|$ | $|\mathcal{E}|$ | $k$ | $d$ | $\sigma_d/d$ | $\hat{\pi}_o$ | Modularity |
|---|---|---|---|---|---|---|---|---|
| DBLP-1 | 213 | 591 | 5489 | 2 | 5.54 | 0.93 | 0.005 | 0.40 |
| DBLP-2 | 237 | 670 | 5935 | 2 | 5.65 | 0.86 | 0.004 | 0.45 |
| FB-1 | 128 | 1593 | 45635 | 3 | 24.89 | 0.44 | 0.055 | 0.53 |
| FB-2 | 621 | 12399 | 718267 | 5 | 39.93 | 0.69 | 0.004 | 0.52 |

Table 2: Summary statistics of the real networks.

| | Louvain $(G)$ | Louvain $(L)$ | Spectra $(L)$ | ORC $(L)$ |
|---|---|---|---|---|
| DBLP-1 | 0.27 | 0.48 | 0.22 | 0.54 |
| DBLP-2 | 0.25 | 0.24 | 0.20 | 0.60 |
| FB-1 | 0.85 | 0.55 | 0.10 | 0.86 |
| FB-2 | 0.56 | 0.31 | $*$ | 0.65 |

Table 3: NMI of different methods on the real data, where "Louvain" represents the Louvain algorithm, "Spectra" represents the Spectral clustering method, "ORC" represents our methods based on ORC, and "$G$", "$L$" indicate that the methods are applied on the original graph $G$ and its line graph $L$, respectively. $*$: this computation did not finish after a day.

As in the synthetic networks, the proposed method via ORC performs well across all data sets; see Table 3. We compare the performance of our approach against other popular community detection methods, specifically the Louvain algorithm (Blondel et al., 2008) and Spectral clustering (von Luxburg, 2007; Damle et al., 2018), both applied to the line graph $L(G)$. Our proposed method via ORC outperforms the reference algorithms most of the time. We emphasize that the proposed method is much more efficient than spectral clustering, with runtime about 1/10 (DBLP-1: 0.02s/run for Louvain $(G)$, 0.36s/run for Louvain $(L)$, 244.42s for Spectral $(L)$, and 35.96s for ORC $(L)$). It is important to note that the performance of the Louvain algorithm depends on the initialization, so multiple runs are recommended for improved performance. Here, we output the average performance from $n_r = 10$ runs. There are various techniques that can be applied to further improve its performance, but these necessarily increase the time complexity (Strehl and Ghosh, 2003). The poor performance of spectral clustering on the line graph (third column of Table 3) should be expected, since unlike in the original graph, the relevant signal is not captured by the top eigenvectors of its adjacency matrix (Lubberts et al., 2021).

## 5. Conclusions

In this paper we have demonstrated the benefits of performing mixed-membership community detection on the line graph rather than the original graph. We used this observation to

propose a curvature-based detection algorithm, which is based on evolving edge weights on the line graph under a discrete Ricci curvature flow. This effectively coarsens the line graph, revealing communities of edges in the original graph, which identifies cut locations between communities even when bridges do not appear. This also results in the identification of mixed-membership vertices that lie between graph communities.

There are numerous avenues for future investigation: The scalability of the algorithm may be boosted by using alternative curvature notions or by avoiding the computation of the full line graph. Another extension would be the incorporation of edge and node weights into the approach. Combining existing bridge-cutting methods with the methods proposed here may lead to clustering that is more robust to different kinds of bridges between communities. Lastly, to evaluate the numerical benefits of the method more comprehensively, a comparison against a wider selection of competing approaches (e.g., Bayesian methods (Airoldi et al., 2008)) and a detailed study of hyperparameter dependencies could be performed.

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

## Appendix A.  Additional Examples

We illustrate the idea of mixed-membership community detection on the line graph with two small examples, one with synthetic data, the other with real data. For the synthetic example, we generate a MMB with two communities ($p_{in} = 0.3, p_{out} = 0$) overlapping in one node (labeled "0") with $X_{0l} = 0.5$, $l \in \{1, 2\}$. Community detection on the original graph $G$ labels node "0" as belonging to one of the communities, whereas community detection on the line graph correctly recognizes the overlap: In Fig. 4, we see that edges that are incident to vertices in each of the two communities are separated.

As a second example, we consider a classic benchmark graph with known community

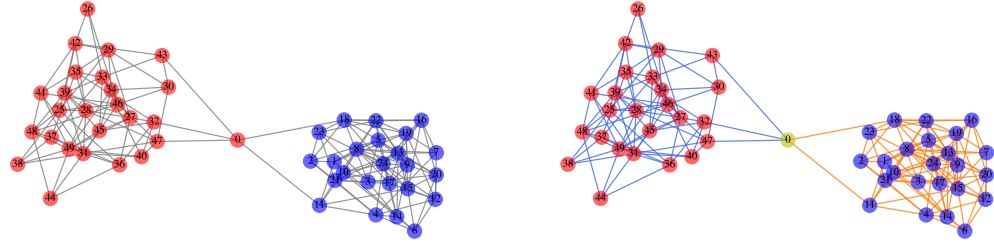

Figure 4:  Communities detected on the $MMB(2, 0.3, 0)$ with node "0" in both communities. Results are shown for community detection on $G$ (left) and $L(G)$ (right). The color of the nodes indicates the community label; intra-community edges are shown in the corresponding color. The results are obtained from $T = 10$ iterations of Alg. 4.1; the threshold $\Delta = 0.1$ was selected following a modularity analysis (see sec. 4.2).

structure, the karate club network. We see in Fig. 5 that community detection on the original graph $G$ mislabels one node in each community. In contrast, community detection on the line graph recognizes that both nodes have a balanced number of connections in each of the communities and can therefore be considered to have mixed-membership.

## Appendix B.  Normalized Mutual Information

For communities $C_1, C_2, \ldots, C_k$, the community membership of each node $i$ can be expressed as a binary vector of length $k$. $(z_i)_l = 1$ if node $i$ belongs to $C_l$; $(z_i)_l = 0$ otherwise. The $l$-th entry of this vector can be viewed as a random variable $Z_l$, whose probability distribution is given by $P(Z_l = 1) = n_l/n$ and $P(Z_l = 0) = 1 - P(Z_l = 1)$, where $n_l = |C_l|$ and $n = |V|$. The same holds for the random variable $Y_h$ associated with another set of communities $C'_1, C'_2, \ldots, C'_{k'}$. Both the empirical marginal probability distribution $P_{Z_l}$ and the joint probability distribution $P(Z_l, Y_h)$ are used to further define entropy $H(\mathbf{Z})$ and $H(Z_h, Y_h)$. The conditional entropy of $Z_l$ given $Y_h$ is defined as $H(Z_l|Y_h) = H(Z_l, Y_h) - H(Y_h)$. The entropy of $Z_l$ with respect to the entire vector $\mathbf{Y}$ is based on the best matching between $Z_l$ and the component of $\mathbf{Y}$ given by

$$H(Z_l|\mathbf{Y}) = \min_{h \in \{1, 2, \ldots, k'\}} H(Z_l|Y_h).$$

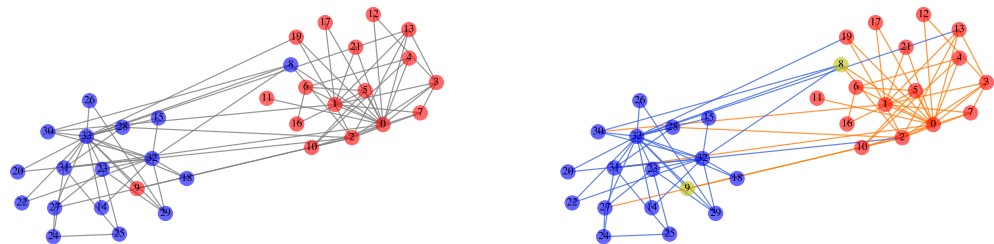

Figure 5: Communities detected on the karate club network. Results are shown for community detection on $G$ (left) and $L(G)$ (right). The color of the nodes indicates the community label, nodes with mixed membership are highlighted in green. Intra-community edges are again shown in the corresponding color. The results are obtained from $T = 50$ iterations of Alg. 4.1; $\Delta$ was again determined from a modularity analysis.

The normalised conditional entropy of $\mathbf{Z}$ with respect to $\mathbf{Y}$ is

$$H(\mathbf{Z}|\mathbf{Y}) = \frac{1}{k} \sum_l \frac{H(Z_l|Y)}{H(Z_l)}.$$

In the same way, we can define $H(\mathbf{Y}|\mathbf{Z})$. Finally, the NMI for two sets of communities $C_1, C_2, \ldots, C_k$ and $C'_1, C'_2, \ldots, C'_{k'}$ is given by

$$NMI(\mathbf{Z}|\mathbf{Y}) = 1 - [H(\mathbf{Z}|\mathbf{Y}) + H(\mathbf{Y}|\mathbf{Z})]/2$$

Hence to use NMI, we convert the label vector $y$ to a binary assignment by first normalising it by its 2-norm and then thresholding its element by $0.8/k$.

