# OpenReview forum: "Mixed-Membership Community Detection via Line Graph Curvature"
_NeurIPS.cc/2022/Workshop/NeurReps — NeurReps 2022 Poster_

### Official Review · Reviewer_y8ix · 2022-10-14
**Incremental work toward the community detection problem**

**Confidence:** 4
**Soundness:** 3
**Presentation:** 4
**Contribution:** 3
**Overall Rating:** 6

**Summary:**

Focusing on the mixed-membership community detection problem, this author argues that the line graph has a unique strength over the original graph. Therefore, this paper proposes a curvature-based method to detect the overlapping community structure. The proposed method adopts Ricci curvature flow to evolve edge weights and thus reinforce community structure for inferencing.

**Questions:**

1. Is the line graph always better than the original for the community detection problem?
The paper uses Figures 1, 4, and 5 to demonstrate that the line graph can capture more information and thus be better for community detection. However, whether such a claim is universal or just works for a few cases is unclear. It would be better if the author could provide more evidence for it.


2. What is the impact of hyper-parameters?
The authors also rely on predefined hyper-parameters in the proposed algorithm, such as $\Delta$. It is unclear to me their impact on the performance.


3. What are the results for “spectra(G)” and “ORC(G)” in table 3?
The author reports “Louvain(G),” “Louvain(L),” “spectra(L),” and “ORC(L)” in table 3 but skips the “spectra(G)” and “ORC(G).” I think it would be better if the author could report them. By comparing different methods on the line graph and original graph, we can understand the necessity of such converting (from original graph to line graph). And the results can also be used to support the previous claims that why we prefer the line graph.


**Limitations:**

1. The time complexity of generating the line graph may not be trivial. On page 9, the authors only report the running time of DBLP-1 (only hundreds of nodes), which is the smallest dataset. Maybe authors should do more work or provide more evidence to show that the proposed methods have practical value.

2. The baseline model is not state-of-art. The Louvain algorithm and spectral clustering algorithm are 15 years old. I encourage authors to compare more state-of-art algorithms. Meanwhile, since the learning-based algorithm has been emerging recently, the authors may also want to emphasize the strength of the proposed algorithm.



**Recommended Decision:**

2: Borderline

**Relevance:**

4: Highly relevant

**Strengths And Weaknesses:**

The paper is well-written and clearly illustrates the problem, motivation, evidence, and solutions. I enjoy reading this paper. The proposed method adopts certain math justifications or follows math intuitions.

The shortcomings mainly come from the rigidity of claims and the completeness of numerical experiments since this is for the proceeding track. The details are listed below.


**Submission Track:**

Proceedings Paper (9 Page)

---

### Official Review · Reviewer_kQPg · 2022-10-15
**Interesting geometric-based method for mixed-membership models**

**Confidence:** 4
**Soundness:** 4
**Presentation:** 4
**Contribution:** 2
**Overall Rating:** 7

**Summary:**

The authors propose a new method to learn a mixed-membership model (i.e. a model that associates each association with multiple clusters via membership probability) from the community graph. To this end, the authors extend the Ricci flow method method proposed by Ni et al. (2019) in the following ways: 1) they apply the method for mixed-membership model instead of clustering and 2) they work on the line graph (the graph of the edges of the original graph) instead of the original graph. Using the normalized mutual information as the performance metric, the experiments on synthetic and real datasets show that the proposed method outperformed previous mixed-membership algorithms.


Reference

Chien-Chun Ni, Yu-Yao Lin, Feng Luo, and Jie Gao. Community detection on networks
with ricci flow. Scientific reports, 9(1):1–12, 2019.

**Questions:**

None

**Limitations:**

The authors have carefully addressed the limitations of their method.

**Recommended Decision:**

3: Accept

**Relevance:**

4: Highly relevant

**Strengths And Weaknesses:**

# Strengths

The authors give a very clear presentation of the subject. For any new geometric object introduced, the authors provide some intuition that makes the material easier to digest. The literature review is very thorough.

# Weaknesses

* Despite the point above, there are some geometric objects that could have been explained better; for example, what are the behaviors of Ricci flow around the regions with positive/negative curvatures? What does it mean for the Ricci curvature to "contract"?
* As mentioned in the conclusion, the proposed method should also be compared to the standard Bayesian method i.e. Airoldi et al. (2008).

Nenetheless, I have learned a lot from reading this work.

Reference

Edo M Airoldi, David Blei, Stephen Fienberg, and Eric Xing. Mixed membership stochastic
blockmodels. Advances in neural information processing systems, 21, 2008.


**Submission Track:**

Proceedings Paper (9 Page)

---

### Official Review · Reviewer_775x · 2022-10-16
**Good paper, but requires clarifications and further evaluation**

**Confidence:** 4
**Soundness:** 4
**Presentation:** 4
**Contribution:** 4
**Overall Rating:** 5

**Summary:**

__Summary.__ The authors consider the problem of detecting mixed-membership communities. They propose an approach based on curvature --- a discrete version of the tools in Differential Geometry used to characterize the local and global properties of geodesic spaces --- which operates directly on the line graph. The idea of geometry-based community detection is to use notions of discrete curvature to partition networks into communities, based on the observation that bridges between communities are more negatively curved than edges within communities.


__Contributions.__ The authors propose a new curvature-based approach applied to the line graph for mixed membership assignment. The authors also do a good job in motivating the curvature-based approach. They also make an interesting argument for using the line-graph, as being more appropriate for mixed-membership detection --- which is not really possible in with the original graph space. So essentially, they transform the problem in attributing labels to edges (eg, edge between community A and community A, edge between community A and B, edge between community B and B), and finding clusters of edges. The hope is that the negative curvature of edges between nodes from different community is bigger than  between edges from the same community.


**Questions:**

See weaknesses

**Limitations:**

The evaluation is fairly limited --- the authors do not give a clear sense of when their method would be advantageous over others.

**Recommended Decision:**

2: Borderline

**Relevance:**

4: Highly relevant

**Strengths And Weaknesses:**

__Strengths__
This approach seems novel, the paper is rather clear, the writing is convincing and the method seems new and promising. This is a good early track paper.


__Weaknesses__
While the authors make a convincing argument for the line graph, it would have been nice to comment on a few peculiarities of this graph ----currently, I wonder if everything follows through, and the authors, who justify the use of Ricci curvature in the original graph, should perhaps detail how these properties are also applicable in the line graph. In particular, the “bridge” nodes in the line graph (ie edges of type “AB”) have stranger properties than the original graph: they don’t cluster (there is no cluster of AB edges), so it’s probably useless to try and apply a clustering algorithm to these. Maybe the geometric approach is indeed better suited in that case – I suppose these edges have negative curvature, because you expect random walks to quickly get away from these nodes as they get stuck in communities of edges of type “AA”, “BB” ---- so maybe the argument that the authors make for the original graph still applies to the line graph.
However, I do not understand (a) how the mixed edges “AB” are taken care of in the mixed membership computation in Eq 4.4; what it means to coarsen the line graph --- what does it mean in the original graph?


**Submission Track:**

Proceedings Paper (9 Page)

---

### Decision · Program_Chairs · 2022-10-21

Accept (Poster)